# Genetic Diversity, Population Structure and Selection Signature in Begait Goats Revealed by Whole-Genome Sequencing

**DOI:** 10.3390/ani14020307

**Published:** 2024-01-18

**Authors:** Haile Berihulay Gebreselase, Hailemichael Nigussie, Changfa Wang, Chenglong Luo

**Affiliations:** 1State Key Laboratory of Swine and Poultry Breeding Industry Guangdong Key Laboratory of Animal Breeding and Nutrition Institute of Animal Science, Guangdong Academy of Agricultural Sciences, Guangzhou 510640, China; 2Department of Biotechnology, College of Natural and Computational Science, Aksum University, Aksum 1010, Tigray, Ethiopia; 3Department of Animal Science, Adigrat University, Adigrat 7040, Tigray, Ethiopia; haylish123@gmail.com; 4Agricultural Science and Engineering School, Liaocheng University, Liaocheng 252000, China; wangchangfa@lcu.edu.cn

**Keywords:** Begait goat, genetic diversity, selection signature, population structure, whole-genome sequencing

## Abstract

**Simple Summary:**

Begait is a goat population reared for meat and milk production in northern Ethiopia, Tigray region. However, molecular-based genetic evaluation of these goat breeds has not yet been performed. Hence, the estimation of their genetic parameters and selection responses is crucial for the conservation and genetic improvement of goats. In this study, we estimated genetic parameters using a whole-genome sequencing data set to analyze genetic diversity, population structure and selection signatures by comparing the Begait goat genome with the remaining four Ethiopian goat populations. Generally, our results indicated the basis for the projecting characteristics, which can be used for future genetic improvement and animal genetic resources of goat breeds.

**Abstract:**

Goats belong to a group of animals called small ruminants and are critical sources of livelihood for rural people. Genomic sequencing can provide information ranging from basic knowledge about goat diversity and evolutionary processes that shape genomes to functional information about genes/genomic regions. In this study, we exploited a whole-genome sequencing data set to analyze the genetic diversity, population structure and selection signatures of 44 individuals belonging to 5 Ethiopian goat populations: 12 Aberegalle (AB), 5 Afar (AF), 11 Begait (BG), 12 Central highlands (CH) and 5 Meafure (MR) goats. Our results revealed the highest genetic diversity in the BG goat population compared to the other goat populations. The pairwise genetic differentiation (F_ST_) among the populations varied and ranged from 0.011 to 0.182, with the closest pairwise value (0.003) observed between the AB and CH goats and a distant correlation (*F_ST_* = 0.182) between the BG and AB goats, indicating low to moderate genetic differentiation. Phylogenetic tree, ADMIXTURE and principal component analyses revealed a classification of the five Ethiopian goat breeds in accordance with their geographic distribution. We also found three top genomic regions that were detected under selection on chromosomes 2, 5 and 13. Moreover, this study identified different candidate genes related to milk characteristics (GLYCAM1 and *SRC*), carcass (*ZNF385B*, *BMP-7*, *PDE1B*, *PPP1R1A*, *FTO* and *MYOT*) and adaptive and immune response genes *(MAPK13*, *MAPK14*, *SCN7A*, *IL12A*, *EST1 DEFB116* and *DEFB119*). In conclusion, this information could be helpful for understanding the genetic diversity and population structure and selection scanning of these important indigenous goats for future genetic improvement and/or as an intervention mechanism.

## 1. Introduction

A goat (*Capra hircus*) belongs to a group of animals called small ruminants and is a critical source of livelihood for rural people [1], thereby contributing to poverty reduction and the means of achieving sustainable agriculture and food security because of its shorter production cycles, greater adaptability to harsh environments and faster growth rates [2]. Goats have been demonstrated to be an important livestock species in the most marginal regions of the globe, due to their versatility domesticates and high adaptive capability to diverse biophysical and production environments from the high intensification in the most developed countries on dairy herds, to the hard conditions of dry regions [3]. They provide multiple economic and social benefits, such as assets (as a form of insurance), small initial investments with quick returns due to fast multiplication, soil fertility improvements and social–cultural and religious aspects of everyday life [4].

Currently, there are more than 1 billion heads of goats around the world, with more than 57.4% and 37.0% found in Asia and Africa, respectively. In developing countries like Ethiopia, enormous goat genetic resources are present [4], among which there are approximately 30.2 million heads of goat population in Ethiopia [5]. These indigenous goat populations have generally developed certain valuable genetic merits, such as the ability to perform better under low input conditions, greater disease tolerance, high adaptive ability to heat stresses and specialized important traits [6]. Moreover, selection for environmental adaptation and marketable traits may have differently shaped the genomes of these goat breeds [7]. Characterizing and understanding genetic diversity and population structure is therefore required to contribute to designing better breeding strategies, allowing sustained genetic improvement, and conserving animal genetic resources to enable genomic selection of goat breeds. The Begait goat is a multipurpose breed found in western Tigray, north Ethiopia. It serves as a significant source of cash income and food, providing both meat and milk [8]. Additionally, the breed is known for its resistance to drought conditions and better milk yield [8,9].

The availability of whole-genome sequencing is an ideal approach to detecting genetic variation, population structure, selective scanning or direct identification of trait-associated sequence variations [10]. By analyzing the whole genomes of goats, researchers can determine genetic diversity, population structure and selection signatures, which can offer insights into designing breeding improvement strategies for indigenous goat genetic resources [11]. This approach also helps in identifying candidate genes for specific traits and local adaptation in goats. However, until recently, only a few studies have been analyzed to explore the genomes of goats. For example, investigations of Ethiopian indigenous goat populations using high throughput sequencing have not yet been reported. In this study, we discovered specific genes that are positively selected in Ethiopian goat breeds associated with meat, milk, carcass and adaptation traits. In addition, we applied fixation index (*F_ST_*) and nucleotide diversity (θπ, pairwise nucleotide variation as a measure of variability)-based cross approaches to explore the genetic variation, population structure and selection signature across the whole-genome sequence. The fixation index (*F_ST_*) measures the proportion of variance in allele frequencies among populations relative to the total variance [10] and is an actual approach to detect footprints for populations with unknown phenotypes under natural or artificial selection, whereas nucleotide diversity (θπ, pairwise nucleotide variation as a measure of variability) is the extent of the proportion of pairwise differences between two populations [12]. Therefore, the purpose of this study was to gain a deeper understanding of genomic diversity and variation related to goat production and reproductive traits, which can contribute to the sustainable utilization and conservation of goat genetic resources. Additionally, the identification of selection signatures and candidate genes can provide a basis for future breeding and conservation efforts.

## 2. Materials and Methods

### 2.1. Sampling Collection, DNA Extraction and Genomic Sequencing

As observed in our previous work [13], DNA was extracted from blood obtained from five goat breeds: 12 Aberegalle (AB), 5 Afar (AF), 11 Begait (BG), 12 Central highlands (CH) and 5 Meafure (MR) goats were sequenced. The five Ethiopian indigenous goat populations live at different altitudes and agro-ecologies, as indicated in Table 1. The geographical locations of all animals used in the present study, specifically Begait goats, are shown in Figure 1. To minimize the number of closely related individuals, animals were selected based on information obtained from livestock owners. Following the procedure of the company (Berry Genomics, Beijing, China), we collected a 5 mL blood sample from the jaguar vein of each goat using an FTA card. The genomic DNA was extracted from the blood using a standard phenol/chloroform extraction method; paired-end sequencing libraries with an insert size of 150 bp were sequenced using an Illumina HiSeq 3000 platform (Illumina, San Diego, CA, USA).

### 2.2. Data Filtering, Clean Read Generation and Variant Discovery

As described in our previous work [13], fastQC software v0.11.5 was used to perform a raw sequence data quality check. The filtered reads were mapped using the ‘mem’ algorithm of BWA24 (v 0.7.12) against the goat reference genome 25 (assembly ARS1). We used the SAMtools package 0.1.2 [14] to convert the SAM file format to BAM and to filter the unmapped reads. On average, 99.8% of the reads were mapped, resulting in a final average sequencing depth of 16 (10× to 30×) per individual (Appendix A). The software Picard (v2.10.6) (http://broadinstitute.github.io/picard/, accessed on 22 February 2020) was used to remove duplicate reads while MarkDuplicates (version 2.1) were used for identifying and tagging duplicate reads in the genomic data, ensuring accurate variant calling and reducing errors in the analysis process, followed by recalibrating base quality scores using GATK26 (v3.7-0) [15]. After mapping, SNP calling was performed using SAMtools version 0.1.2 [14], and the output was further filtered using the software VCF tools version 0.1.16 [16]. We used the following SNPS exclusion criteria: (1) 3× < mean sequencing depth (overall included individuals) <30×; (2) a minor allele frequency >0.05; (3) maximum missing rate <0.1; and (4) only biallelic autosomes were used for further analysis.

### 2.3. Genetic Diversity, Population Differentiation and Population Structure Analysis

To investigate genetic diversity among the five local breeds, we calculated the proportion of polymorphic loci (P_N_) and observed (Ho) and expected (He) heterozygosity using the—hardy flag, whereas the inbreeding coefficient (*F_IS_*) and minor allele frequency (MAF) were calculated using the commands—het and—frq, respectively, using Plink [17]. We inferred a population-level phylogenetic tree using MEGA X [18]. Itol (http://itol.embl.de, accessed on 22 February 2020) was used to visualize the phylogenetic tree. Principal component analysis (PCA) was used using PLINK 1.9 [17] and plotted in R package version 3.4 [19]. PCA was performed after filtering out SNPs with MAF < 0.05 and a genotype missing rate of less than 90%. ADMIXTURE version 1.3.0 was used to assess the population genome-wide genetic structure among the populations [20]. We further run for each possible group number (K = 2–5) to explore the convergence of individuals. To determine the genetic differentiation among the goat populations, we measured pairwise *F_ST_* values [21]. Before constructing genetic relationship and structure analyses, the total biallelic SNPs were pruned based on linkage disequilibrium (LD) using the ‘–indep-pairwise 50 10 0.2′ command in PLINK v1.9 [17].

### 2.4. Genomic Linkage Disequilibrium (LD) and Effective Population Size (Ne)

Pearson’s squared correlation coefficient (*r*^2^) was used to estimate the LD value of all paired SNPs. The *r*^2^ was detected apart from 20 SNPs within a minimum and 1 Mb maximum distance [16]. Linkage disequilibrium (LD) was calculated using PopLDdecade v3.40 [22] software from the vcf file format. The LD decay between SNP pairs was plotted based on the average r^2^ values for each 20 kb using R software version 3.4 in PopLDdecay [22]. Effective population size (*Ne*) was estimated using SNeP (v1.1) [23] with default parameters. The SNeP was used to estimate each population before 1000 generations using the equation E (*r*^2^) = (1 + 4N_e_c)^−1^ [24], where Ne is the effective population size and c is the recombination rate in cM with 1 cM equal to 1 Mb [16].

### 2.5. Detection of the Genomic Regions under Selection

We used fixation index (F_ST_) and nucleotide diversity (θπ, pairwise nucleotide variation as a measure of variability) statistical methods for the detection of selection signals. To identify candidate loci subjected to selection, we performed selection mapping to explore differences among Ethiopian goat populations. Accordingly, we compared the genome of BG goats (used as a test population) versus the control group (AB, AF, CH and MR goats grouped into one population). We select the region showing high F_ST_ values and low levels of θπ values. The F_ST_ values were then averaged over SNPs using a 100 kb sliding window (≥50 SNP) with a 50 kb step size as described previously by [14]. We then *Z* transformed the F_ST_ values, and θ_π_ ratios were log_2_ transformed. We used the windows with the top 5% values for Z (F_ST_) and log_2_ (θ_π_ ratio) simultaneously as the candidate outliers under strong selection signals. All of the outlier windows were assigned to the corresponding SNPs and genes. Similarly, VCFtools [16] was used to calculate Tajima’s D statistic to confirm the exact selective signals.

### 2.6. Functional Analysis of Candidate Genes under Selection

We performed Gene Ontology (GO) and pathway analyses with DAVID web server 6.7 (http://david.abcc.ncifcrf.gov, accessed on 22 February 2020) and an annotation tool for gene enrichment analysis to further understand the biological functions and pathways of selected genes. To annotate SNPs with respect to their functional classification of genes, we downloaded the caprine gene (CAPRIN2) from the Ensemble (http://www.ensembl.org/, accessed on 22 February 2020) databases, and R software (version 3.4.1) was used for hierarchical clustering of GO terms from DAVID. We then submitted to DAVID Bioinformatics resources for enrichment analysis of the significant overrepresentation of GO biological processes (GO-BP), molecular function (GO-MF), cellular component (GO-CC) and KEGG pathway. Only pathways or annotations with *p* < 0.05 were used. The candidate genes and pathways associated with milk and meat traits were manually selected based on their biological function and a literature search in PubMed.

## 3. Results

### 3.1. Overviews of Genetic Diversity

In the present study, we generated a total of 3.801 GB of data with an average depth of 16× from 44 total samples ranging from 10× to 30× (Appendix A). The transition/transversion ratio was 2.382, which is very similar to the results obtained for other goats [25]. After filtering the processes, 16,439,441 loci were found to be polymorphic. A total of 3,377,527 SNPs were removed, leaving 13,061,914 for further analysis. A significant deviation for 2,822,171 SNPs was excluded from the MAF < 0.05. Approximately 73,600 and 481,792 variants were removed from the Hardy–Weinberg equilibrium (1 × 10^−5^) and call rate < 0.90, respectively. The numbers of variants in each goat are shown in Table 2. CH goats (12,229,657) had the highest number of SNPs, followed by AB (11,137,576), BG (10,833,847) and AF (10,760,581). The lowest numbers of SNPs (10,749,996) were displayed in the MR goat population. To understand the genetic diversity among the 5 goat breeds, we examined the proportion of polymorphic loci (P_N_), expected heterozygosity (H_E_), observed heterozygosity (H_O_), minor allele frequency (MAF) and inbreeding coefficient (Table 2). The proportion of polymorphic loci (P_N_) was found to be lowest in MR (78.58%) and highest in BG (85.42%). In the present study, expected heterozygosity values were greater than the observed heterozygosity (H_E_ > H_O_) at all loci. The AF goat (H_O_ = 22.8%) and MR goat (H_O_ = 26.3%) had the lowest observed heterozygosity, while the BG goat (H_O_ = 35.2) had the highest observed heterozygosity. The average MAF was not significantly different among all the populations. The BG, AB and CH goats had positive inbreeding (F_IS_ = 0.01, 0.03 and 0.04) compared to the other populations, respectively. We also measured the GC percentage across the genome distribution and discovered 43% (Appendix A).

### 3.2. Population Structure Analysis

To assess the genetic relationships and structure among the local breeds, phylogenetic tree, ADMIXTURE and principal component analysis (PCA) were performed. Based on autosomal SNPs, we constructed a phylogenetic tree and demonstrated that BG goats clustered independently, whereas the four goats were grouped into one cluster (Figure 2a). The ADMIXTURE analysis revealed that ranging from K = 2–5, we obtained the most likely biological interpretation and minimum crossover error at K = 2 (Figure 2b). Interestingly, it confirmed that BG goats had an independent admixture pattern compared to the other populations. The first eigenvector (PCA1) explained 2.37% of the total genetic variation and separated the BG from the other (AB, AF, CH and MR) goat populations, which is consistent with the results of the neighbor-joining tree, reflecting their different genetic compositions (Figure 2c). The second eigenvector (PCA2) accounted for 1.76% of the total variance, and relatively close relationships were observed between the AB, CH and MR goats. Close genetic relationships were observed between the CH and AB goats, which is consistent with the population structure. However, a considerable genetic difference was distinguished in the BG goat populations, reflecting their different genetic compositions.

### 3.3. Genetic Relationship between Populations

To determine the extent of genetic divergence between the study goat populations, pairwise genetic differentiation (F_ST_) and Royelonds’ genetic distance were measured for all the autosomal informative SNPs (Table 3). Among the five goat populations, the level of genetic differentiation over the whole genome was low (*F_ST_*= 0.02) but significant (*p* < 0.001) and varied from 0.011 to 0.182, with the closest pairwise value (0.003) observed between AB and CH goats and a distant relationship (*F_ST_* = 0.182) between AB and BG goats.

### 3.4. Genomic Linkage Disequilibrium (LD) and Effective Population Size (Ne)

Information on genome-wide linkage disequilibrium (LD) and effective population size (*Ne*) contains relevant genomic information for measuring genetic diversity in all livestock populations, including goats. Or result clearly showed the LD and *Ne* of each Ethiopian goat population. The analysis revealed that the average LD (*r*^2^) of all populations decreased rapidly from 0–0.5 Mb. In this regard, the estimated *r*^2^ values were corrected with the sample sizes. As a result, AB, BG and CH goats indicated an overall slow decay rate and highest average LD (*r*^2^), whereas the results indicated that the average LD (*r*^2^) for AF and MR goats was the lowest level of LD and decayed significantly faster than that of the other populations (Figure 3a). The *Ne* declined between 3000 and 983 generations ago across the five studied goat populations (Figure 3b). The estimated *Ne* for the AB, CH and BG goats was similar and higher than that for the MR and AF goats 983 generations ago.

We next performed the distributions and the total numbers of SNPs per chromosome (CHI) in all the present studies of goat populations are summarized in Figure 4. Subsequently, CHI-1 contained the highest number of variants (86,2218), with a length of 157.4 Mb, and CHI-25 (203,338), with the shortest length of 42.86 Mb. In accordance with our results, genes related to meat, milk production, coat color and adaptive performance were under selection in the Begait goat populations.

### 3.5. Detection of Selection Signatures

Based on the results of the PCA and population structure analysis, we calculated the fixation index (*F_ST_*) and nucleotide diversity (*θ*_π_ ratio) with a 100 kb sliding window and 50 kb sliding step using a top 5% threshold outlier window. To detect the signature of selection, we compared the genomes of BG goats (high production performance) with the genomes of the control groups (AB, Afar, CH and MR grouped together into one population). The Manhattan plot of the transformed Z (*F_ST_*) and log_2_ (*θ*_π_ ratio) values are presented in Figure 5a,b. We found that the top regions are located at 11.9–11.20 Mb on chromosome 2 (Z*F_ST_* = 14.13, *F_ST_* = 0.45 and log_2_ (*θ*_π_ ratio) = 1.46)), 25.20–25.40 Mb on chromosome 5 (*ZF_ST_* = 4.36, *F_ST_* = 0.16 and log_2_ (*θ*_π_ ratio) = 1.37)), 16.50–16.70 Mb on chromosome 13 (*ZF_ST_* = 6.03, *F_ST_* = 0.21 and log_2_ (*θ*_π_ ratio) = 1.64)) and a negative value of Tajima’s D (Figure 5d). This suggests that the BG goats were considered to be the specific region under positive selection.

Furthermore, using web-based DAVID Bioinformatics resources (v. 6.8, https://david.ncifcrf.gov/, accessed on 1 April 2021), we annotated the top 5% of *F_ST_* values and *θ*_π_ ratio cutoffs (Z (*F*_ST_) > 1.780 and log_2_ (*θ*_π_ ratio) > 0.308), and we identified ~732 (Appendix A) and ~727 (Appendix A) outlier windows as selection regions, respectively. Moreover, we identified 11 candidate genes that are related to growth, milk, defensive mechanisms, melanogenesis and heat tolerance traits (Table 4).

Only the overlapping windows that were detected via both approaches were further considered putative selection signatures to improve the confidence for the identified selection signature. Accordingly, 384 windows under selective signals were common for both of the statics (Figure 6). The functions of the candidate genes were consulted based on the annotations in the NCBI (http://www.ncbi.nlm.nih.gov, accessed on 1 April 2021) and Ensemble (http://www.ensembl.org/, accessed on 1 April 2021) databases. The genes were associated with diverse biological functions, and some had roles in multiple functions.

A total of 36 GO terms (17 in biological processes, 12 in cellular components and 7 in molecular functions) and 6 KEGG pathways were annotated (Appendix A). The overrepresented biological process (BP) terms were related to angiogenesis, positive/negative regulation processes, signaling pathways and metabolic processes. The cellular components (CC) included the centrosome, membrane, chromosome and cell surface. The molecular functions (MF) were related to enzyme, protein, zinc and oxidative activities. Adrenergic signaling in cardiomyocytes, amyotrophic lateral sclerosis, the renin–angiotensin system and the GnRH signaling pathway were enriched in the KEGG pathway (Table 5). In this study, we mainly focus on and discuss the genes and pathways that putatively contribute to the economically important traits of Begait goats.

## 4. Discussion

### 4.1. Genetic Diversity Characteristics

The study of genetic diversity within and across breeds provides insight into population structure and relationships and is essential for the conservation of populations so that populations can face environmental challenges in the future and can respond to long-term selection, either natural or artificial, for traits of economic and cultural interest. The main objective of the present study was to characterize the genetic diversity and population structure and detect signatures of selection in five Ethiopian goat populations using whole-genome sequencing data. Based on the population structure and genetic differentiation analyses, low to moderate genetic differentiation (*F_ST_*) between the five Ethiopian goat populations was observed. The potential reason for this low genetic differentiation suggests high historical gene flow between the goat populations and a recent common history of ancestry or admixture, which was likely related to the historical movement of goats across and the husbandry system for searching for the availability of feed and water. Using the whole genome data set, [26] reported low to moderate genetic similarity for Sudan’s goat population. Populations with a high proportion of polymorphic SNPs indicate segregation of SNPs [27]. We examined the proportion of polymorphic loci among the five Ethiopian goat populations and ranged from 78.58% to 85.42%, which is consistent with the previously reported values of 84.3% to 94.0% for different Chinese goats [28], 88.5% to 92.8% for Ugandan goat populations [29], 84.22% to 97.58% for South African goats [30] and 88.07% for South African Angora goat breeds [31]. However, our results were lower than the 99.5% and 99.70% reported for the Australian [27] and Italian goat populations [32], respectively.

Heterozygosity is a critical measure to understand biological diversity in goat populations. The analysis of diversity within these five goat populations showed that the BG (H_E_ = 60.7%; H_O_ = 35.2) had the highest level of diversity, while the AF (H_E_ = 26.6; H_O_ = 22.85%) and MR (H_E_ = 29.7%; H_O_ = 26.3%) demonstrated the lowest level of genetic diversity. The heterozygosity values found in the present study were higher than the previously reported heterozygosity (H_O_ = 17.2%) for Moroccan goats [33]. However, the results presented in this study are in agreement with those of (H_O_ =23.5% for Bange) and (H_O_ = 24.0%) for Chaidamu Chinese goat breeds [28]. Moreover, very similar to our results, heterozygosity (H_E_ = 37.7%) for Kigazi Ugandan goats [34], H_E_ = 41% for Xhosa ecotype South African goats [30] and H_E_ = 37% for Valdostana Italian goats [32] were reported. The results indicated that the BG goat breeds had the highest expected heterozygosity. This might be explained by the types of crossbreeding programs practiced by farmers keeping BG breeds, resulting in an admixed population. In Ethiopia, specifically the western Tigray region, organized breeding strategies using artificial selection are practiced for BG goat breeds that have been reared for a long time in separate geographic locations under the government breeding research station, resulting in genetic variability. The lowest values observed heterozygosity for the Afar and MR goat breeds might be clarified by small sample size, artificial selection and inbreeding compared to other populations. The observed higher variability between the studied populations could possibly be attributed to a lack of strong artificial selection pressure and a high level of genetic admixture [35]. The observed heterozygosity was lower than expected (H_O_ < H_E_) for all populations, showing a departure from the Hardy–Weinberg equilibrium (HWE) and potentially attributing the discrepancy to forces such as selection against heterozygotes or inbreeding; for example, a positive inbreeding coefficient was observed for AB, BG and CH, which might be due to the whalund effect rather than to inbreeding [36]. These results were consistent with previous findings for Nubian goats (*F_IS_* = 0.001) in Sudan [26], South African, French and Argentinian goat populations (*F_IS_* = 0.009) [37] and for goat breeds in Portugal and Brazil (*F_IS_* = 0.05) [36]. Our results also showed a slight negative inbreeding coefficient for MR and AF goats, suggesting a small sample size. Hence, a larger sample size would be needed to obtain a better estimate of inbreeding measures. Several inbreeding coefficients reflect more distant inbreeding, while others reflect more recent inbreeding [27]. When there is more recent inbreeding in a genome, there would be less distant inbreeding in that genome, causing a negative correlation. Furthermore, our results indicated that the CH and AB goats present similar genetic diversity estimates, which is probably due to similar population management practices and recent isolation among the populations.

As explained in our previous study [13], the whole genome data set confirmed weak phylogenetic relationships and principal component analyses, which indicated closer relationships among the AB, AF, CH and MR goat populations but independent clustering for the BG goat breeds [13]. This result was further confirmed by population admixture analysis, which indicated that some signals of admixture and genetic relationships existed between the populations. The lack of genetic separation among the AB and CH breeds might indicate a high level of genetic similarity and low discrepancy, which may occur as a result of gene flow among the AB and CH breeds. Common ancestry, short domestication history, lack of selection pressure and movement of goats may play a role in the lack of genetic separation in diverse geographically separated populations. Similarly, MR and AF breeds are to some extent clustered together, which may be a result of the movement of animals between the communities in those two regions due to the aforementioned cultural ceremonies and geographically near linkages. Moreover, at K = 2 and 3, the BG goat showed a separate cluster compared to the remaining populations. At K = 4 and 5, all the populations showed a high degree of admixture. This finding could be explained by the fact that BG goats are geographically isolated breeds and have a long domestication history. Furthermore, it will be of interest to expand this breed-level analysis in subsequent studies through the inclusion of more goat samples from its breed, including from Ethiopia and other countries in Africa, Asia and Europe, to better understand the links between breeds.

With the advent of high-density SNP chips and high-throughput genotyping technologies, LD and *Ne* provide substantial information to study genetic diversity, past/recent events and potential responses to both natural/artificial selection in various livestock species, such as cattle [38], sheep [39,40], goats [30] and pigs [41]. Both of these parameters are crucial parameters and powerful methods to characterize and understand the genetic architecture underlying complex traits [38]. We investigated high levels of LD for the AB, BG and CH breeds and slow decay rates, whereas the breeds from MR and AF exhibited a rapid decay rate and a low level of LD. Similarly, AB, CH and BG goat breeds displayed higher *Ne* at 983 generations ago, suggesting that these animals could have been influenced by artificial selection and probably due to the upward selection intensity, the higher sample size and the high inbreeding coefficient, whereas a decreasing recent *Ne* was observed for the MR and AF breeds, signifying that these animals were subjected to genetic drift that resulted in decreased population size. This result is consistent with that reported by [42] for the Kenya Tankwa ecotype and commercial goat breeds across generations, which detected higher effective population sizes. Our results revealed the highest and lowest number of variants detected on chromosomes 1 and 25, respectively. Similar trends were reported for nine Canadian goat populations [43]. This can be attributed to the sizes of the two chromosomes, with chromosome 1 (154.929 Mb) being the longest and chromosome 25 (41.478 Mb) being the shortest [43].

### 4.2. Selection Signature and Identification of Candidate Genes

In domestic animals, detecting evidence of recent positive selection signatures can provide information on genomic regions that are under the influence of both artificial and natural selection and can help to identify beneficial mutations and underlying biological pathways (regions of interest in the genome) for economically important traits. Furthermore, the identification of genes that have undergone positive selection is also an important step for understanding how populations have adapted to environmental changes. Therefore, to identify production and adaptive characteristics, extensive investigations on selection signatures have been carried out among various goat breeds. In this study, we used two different yet complementary statistical approaches, fixation index (*F_ST_*) and nucleotide diversity (*θ*_π_ ratio), which are extensively used in identifying selection signatures. The *F_ST_* statistic and nucleotide diversity (*θ*_π_ ratio) appear to be the most popular choices for detecting signatures of selection [44]. The *F_ST_* can be used primarily to assess the difference in allele frequencies between populations and determine how divergent selection may have affected the genomic pattern of these populations to infer the selective pressure in one population relative to the other [45]. Moreover, Tajima’s D statistic (Tajima, 1989) provides a popular method for identifying specific regions in a genome. Different authors have used Tajima’s D to detect evidence (confirmation) of positive selection in sheep [46] and goat populations [44]. Previously, these methods were successfully applied in cattle [47], sheep [46], goats [44] and chickens [48]. Using these approaches, we identified several genes and genomic regions under positive selection related to body weight, growth rate, fat deposition, carcass, meat, milk and adaptive and immune response-related traits.

Goats are among the principal meat- and milk-producing animals and make immense contributions to poor livelihoods. In Ethiopia, particularly in the Tigray region, Begait goats, Begait cattle and Begait sheep are among the three major livestock species that serve as sources of meat and milk in the northern region of Ethiopia [9,49]. Most of the time, tropical breeds tend to have small body weights/sizes and growth rates compared with temperate breeds [50]. Hence, natural selection may have left genomic footprints in the underlying genes involved in the production traits of these goats. We identified candidate genes associated with milk and carcass traits (*ZNF385B*, *BMP-7*, *PDE1B*, *PPP1R1A*, *FTO* and *MYOT*), which supports this hypothesis. *ZNF385B* is implied as a candidate gene for obesity in humans and pigs [51]. The *BMP*-7 gene has an essential biological function in embryonic development, growth and cellular functions [52,53]. It also plays a critical role in the regulation of cartilage development, maintenance and repair [54]. The *PDE1B* gene is highly associated with fat deposition and carcass traits in Korean cattle [55]. *PPP1R1A* was observed to be significantly associated with body mass index (BMI) [56], and it is also known to be positively correlated with insulin secretion in humans [57]. The *FTO* gene is strongly associated with carcass quality traits in cattle [58], and it also affects body mass index (BMI) in humans [59]. The other gene of interest that we identified is *MYOT*, which is associated with myofibril assembly and actin binding in muscle tissue [60]. Recent functional analysis revealed the association of mutations within *MYOT* with loin muscle area, backfat thickness and intramuscular fat in cattle [61]. Milk production and composition traits are under the control of multiple genes and are other economically important traits in dairy goat breeds. More interestingly, we detected *GLYCAM1* and *SRC* genes, which showed evidence for positive selection in milk quality traits. The gene *GlyCAM1 is* known as cell adhesion molecule 1 (*GlyCAM1*), which has been shown to be highly expressed in pregnant and lactating mammary glands in goats [62] and bovines [63]. In our study, we detected high evidence of higher Z (*F_ST_*) and log_2_ (*θ*_π_ ratio) values but lower Tajima’s D values for the gene *GLYCAM1* compared with those in the adjacent genomic regions, indicating that a strong selective sweep occurred in this gene. The *SRC* gene appears to be essential for increased expression of the prolactin receptor and alveolar cell organization [64], and it is the direct regulator of the transcription/translation of milk protein genes in mammary epithelial cells in mice [65].

The indigenous Begait goat is also known to be well adapted to a wide range of stresses (feed and water shortages, high ambient temperature and adaptive immunity) and has evolved to survive and reproduce under such conditions, and its genome should harbor footprints for adaptation. Coat color is an important trait in locally adapted goat breeds. It is an important feature of the goat for determining the radiant heat load as to how much solar radiation is reflected from their body and how much is absorbed. Animals with light coatings absorb less heat than those with darker coatings [66]. Begiat goats are predominantly white coat color and hairy thighs with long drooping ears [49]. In this study, we identified genes (*SLC6A2*, *SLC7A11*, *SLC26A8*, *ATP6V1H*, *MAPK13*, *MAPK14*, *SCN7A*, IL12A, EST1 *DEFB116* and *DEFB119*) that contribute to adaptive and immune responses.

The *SLC7A11* and *SLC26A8* genes are related to coat color. The *SLC6A2* gene is associated with an adaptive response in bovines [67]. Mutation in *SLC7A11* implied coat color formation in rabbits [45]. The ATP6V1H gene is a member of the family of vacuolar ATPases (V-ATPases), which play a vital role in dry land adaptation in goats [68] and Buffalo [69]. The *MAPK13* and *MAPK14* genes are involved in different aspects of thermotolerance. Chronic pain is considered a major physical and mental health problem in animals. The positive selection of genes involved in gene ontologies and pathways related to the immune response has been previously reported for African goats [70]. The interleukin 12A (*IL12A*) gene is a family of cytokines that may be associated with the immune response [29], and *EST1* is another gene responsible for the regulation of immune cell function [71]. The *DEFB116* and *DEFB119* genes are associated with the defense response [72].

## 5. Conclusions

In this study, we found low to moderate genetic diversity indices. Moreover, to detect selective signatures, we used two complementary methods (Z*F_ST_* and log_2_ (*θ*_π_ ratio)) across the whole genome of the five indigenous goats using whole genome sequencing data. Our analysis revealed multiple candidate genes under positive selection, which are related to milk production, carcass and adaptive traits. Interestingly, we identified the *GLYCAM1* signaling pathway, which is an intriguing candidate pathway. Hence, our results can contribute to the identification of the variants that underlie genetic diversity, population structure and the detected selection signatures. In most cases, further studies are required to further confirm and refine our results by integrating comprehensive genomic data.

## Figures and Tables

**Figure 1 animals-14-00307-f001:**
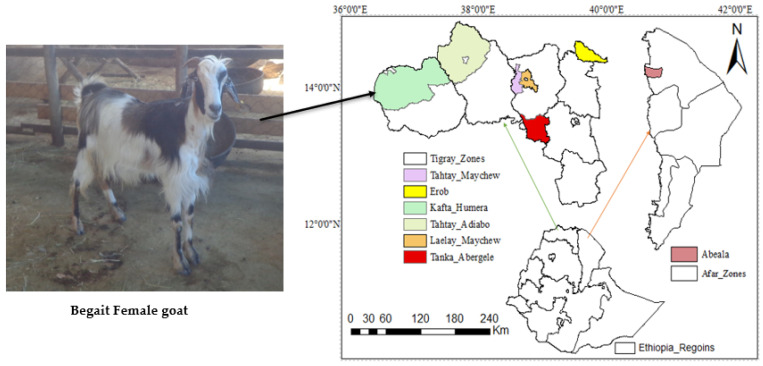
Geographic distribution of the five Ethiopian goat populations.

**Figure 2 animals-14-00307-f002:**
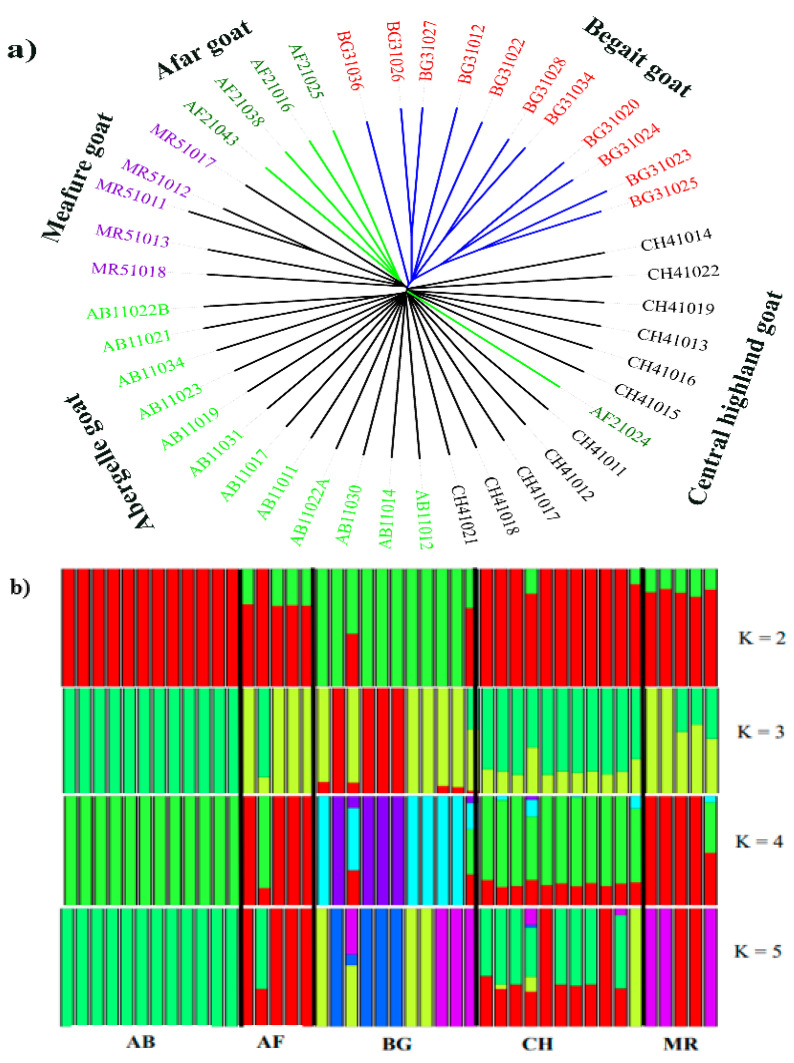
Population genetic relation analysis in five indigenous goats. (**a**) Phylogenetic tree. (**b**) Population structure. (**c**) Principal component analysis.

**Figure 3 animals-14-00307-f003:**
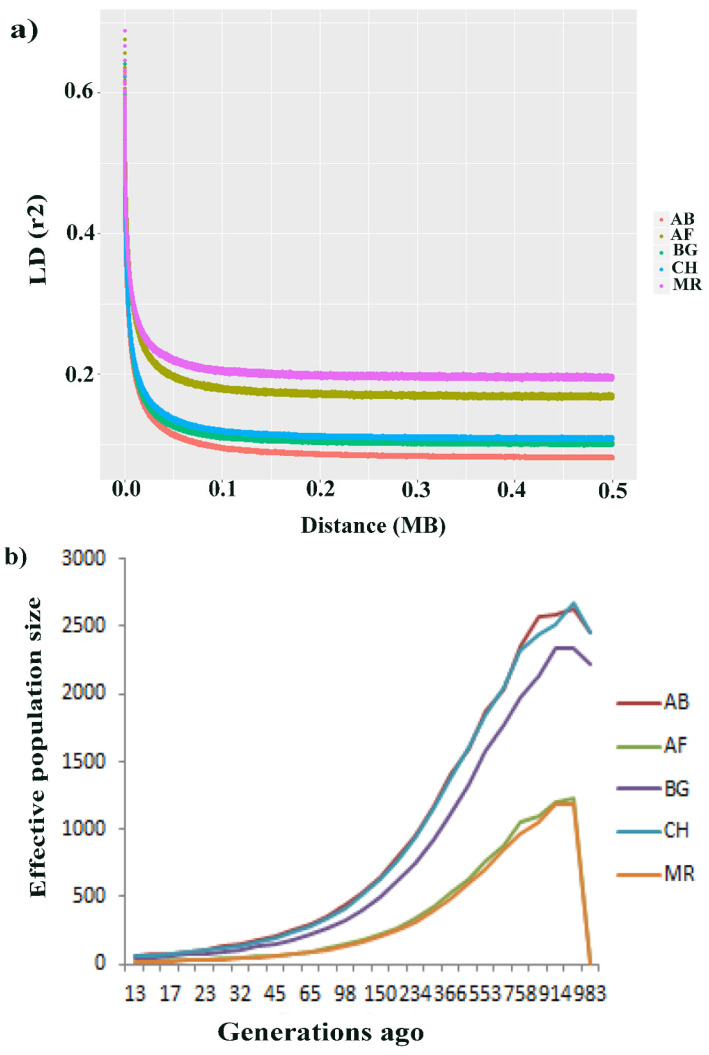
Average estimated linkage disequilibrium and effective population sizes in five goat populations. (**a**) LD decay (r^2^) from 0 to 0.50 mb. (**b**) *Ne* over the past 983 generations.

**Figure 4 animals-14-00307-f004:**
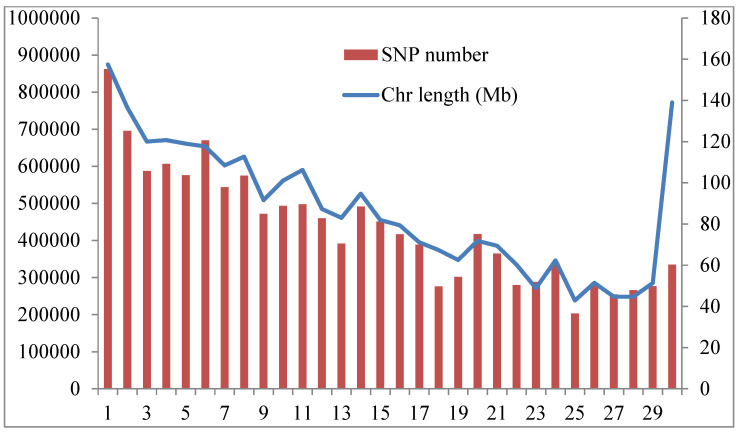
Summary of SNPs and chromosome lengths included in the analysis. Red bars indicate the number of SNPs, and the blue line indicates chromosome length.

**Figure 5 animals-14-00307-f005:**
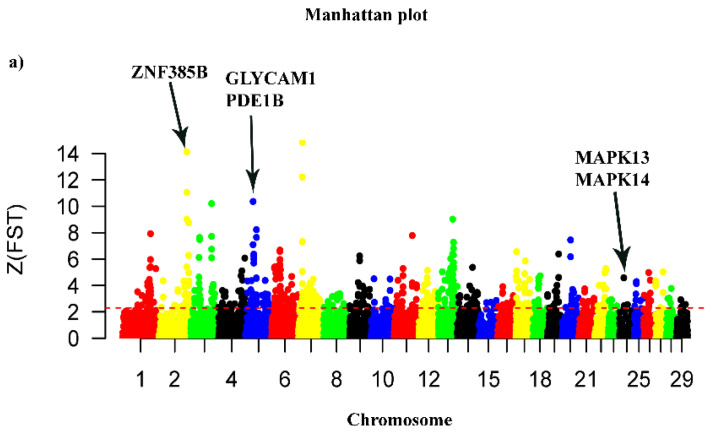
Detection of selection signals in Begait goats. (**a**) Manhattan plots of ZF_ST_. (**b**) Manhattan plots of log2 (θπ—control/θπ—Begait goat). The ZF_ST_ and log2 (θπ ratio—control/θπ ratio—Begait goat) were calculated in 100 kb windows sliding in half steps using the top 5% thresholds. (**c**) F_ST_ and log_2_ (θ_π_ ratio) values around the GLYCAM1 locus. The yellow and gray lines represent the F_ST_ and log_2_ (θ_π_ ratio) values, respectively. (**d**) Tajima’s D values around the GLYCAM1 locus. The yellow and gray lines represent the Begait and control goats, respectively.

**Figure 6 animals-14-00307-f006:**
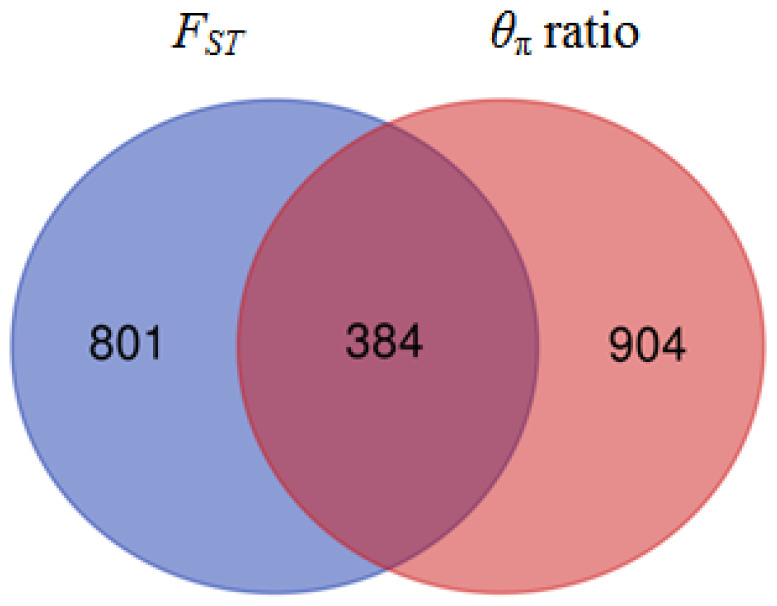
A Venn diagram overlapping the outlier plot of *F*_ST_ and π ratio.

**Table 1 animals-14-00307-t001:** Goat populations, regions of sampling locations, altitudes, GPS coordinates and agro-ecologies.

No	Goat Breeds	Location	Altitude (m.a.s.l)	GPS Coordinator		Classification of Agro-Ecologies
				Latitude	Longitude	
1	Begait	Tigray region (Western zone)	560–1849	13°14′–14°27′ N	36°27′ E–37°32′ E	low altitude
2	Aberegalle	Tigray (Central zone)	1200–1500	13°19′60.00′ N	38°49′59.99″ E	low altitude
3	Afar	Afar (Abeala)	100–1500	13°21′23.05′ N	39°45′27.10″ E	low altitude
4	Central highlands	Tigray (Central zone)	1500–2300	14′0715.92′ N	38°43′24.13′ E	mid altitude
5	Meafure	Tigray (Eastern zone)	1500–2300	11°23′–11°44′ N	38°20′–38°44′ E	mid altitude

**Table 2 animals-14-00307-t002:** Genetic diversity of the five goat populations.

Population	*n*	SNPs	PN	H_O_	H_E_	MAF < 0.05	F_IS_
Abergelle	12	11,137,576	82.69	32.2	34.5	0.222	0.03
Afar	5	10,760,581	78.62	22.8	26.6	0.214	−0.04
Begait	11	10,833,847	85.42	35.2	40.2	0.221	0.01
Central highland	11	12,229,657	82.52	30.7	30.8	0.224	0.04
Meafure	5	10,749,996	78.58	26.3	29.7	0.223	−0.18

Note: PN = proportion of polymorphic loci; H_O_ = observed heterozygosity and H_E_ = expected heterozygosity; MAF = minor allele frequency and F_I_s = inbreeding coefficient.

**Table 3 animals-14-00307-t003:** Pairwise genetic differentiation (*F_ST_)* (below diagonal) and Reynolds’ genetic distance (above diagonal) among five Ethiopian indigenous goat populations.

Breed Name	AB	AF	BG	CH	MR
AB		0.107	0.065	0.151	0.078
AF	0.014		0.105	0.036	0.123
BG	0.182	0.019		0.15	0.078
CH	0.003	0.006	0.027		0.16
MR	0.019	0.007	0.028	0.011	

**Table 4 animals-14-00307-t004:** The candidate genes identified in Begait versus the control group comparisons.

Traits	Chr.	Position (Mb)	Candidate Genes	Z (*F*_ST_)	log_2_ (*θ*_π,_ Ratio)
Defensive mechanism	13	60.30–60.50	*DEFB116*, *DEFB119*, *DEFB124*	6.37	0.55
Melanogenesis	13	59.70–59.90	*SLC52A3*	3.65	2.38
	13	25.20–25.40	*SLC6A2*	4.11	0.31
Growth and milk quality traits	5	25.20–25.40	*GLYCAM1*, *PPP1R1A*	4.36	1.37
	18	23.90–24.10	*FTO*	3.83	0.32
	13	65.80–66.00	*SRC*	2.62	0.36
Heat tolerance	23	38.60–38.80	*MAPK13*, *MAPK14*	2.45	0.75

**Table 5 animals-14-00307-t005:** The Kyoto Encyclopedia of Genes and Genomes (KEGG) pathways obtained from DAVID Gene Ontology analysis using the Z (*F_ST_*) and log_2_ (*θ*_π_ ratio) combined gene list.

Category	Term	*p* Value	Genes
KEGG_PATHWAY	hsa04261: Adrenergic signaling in cardiomyocytes	0.001	*AGTR1*, *MAPK13*, *MAPK14*, *BCL2*, *PPP1R1A*, *CACNB1*, *CAMK2D*, *SCN7A*, *CACNA2D2*
KEGG_PATHWAY	hsa04912: GnRH signaling pathway	0.01	*MAPK13*, *MAPK14*, *CAMK2D*, *PRKCD*, *MMP2*, *SRC*
KEGG_PATHWAY	hsa05152: Tuberculosis	0.02	*MAPK13*, *MAPK14*, *BCL2*, *IL12A*, *CAMK2D*, *ATP6V1H*, *EEA1*, *SRC*
KEGG_PATHWAY	hsa05014: Amyotrophic lateral sclerosis (ALS)	0.04	*GRIN2B*, *MAPK13*, *MAPK14*, *BCL2*
KEGG_PATHWAY	hsa05169: Epstein–Barr virus infection	0.04	*ITGAL*, *PSMD12*, *MAPK13*, *MAPK14*, *BCL2*, *PSMD2*

## Data Availability

The whole-genome sequencing data are deposited and available at http://bigd.big.ac.cn/gvm/getProjectDetail?project=GVM000049; accession number: GVM000049.

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
