# Peer review of "Genetic Diversity, Population Structure and Selection Signature in Begait Goats Revealed by Whole-Genome Sequencing"

_animals, 2024, doi:10.3390/ani14020307_

Round 1

Reviewer 1 Report

Comments and Suggestions for Authors

In the present manuscript the authors perform a molecular-based genetic evaluation (by whole genome sequencing of 44 samples) of five Ehiopian goat breeds, with a major focus on the Begait breed. They compare the data and find information about genetic diversity, differentiation among populations and genomic regions under selection. The topic is of interest to the field, and the manuscript overall is well written, but it cannot be published as it is.

What was written in the introduction is fine, but a description of the breeds studied in their habitat is missing (at least provide some references) and a hypothesis and purpose of the work, at the end of the Introduction section, are missing. 

Many references are quite old, recent publications are missing (only 2 from 2023). In my opinion it is fine to cite the first studies that explored a topic, but it is also necessary to cite the most recent studies because they certainly bring an update to the subject and new points of view.

I inserted the remaining comments in the .pdf of the manuscript.

Comments on the Quality of English Language

English is fine for me, the manuscript is easily readable, there are some minor typo, which I indicated in the .pdf file

Author Response

Dear respected reviewer,

Thank you for your valuable feedback on the title "Genetic Diversity, Population Structure and Selection Signature in Begait Goats Revealed by Whole-genome Sequencing." We appreciate your suggestions and have made revisions to address your concerns. The following is a summary of the changes made based on your feedback:

  • We have rephrased the title to make it more concise and informative, while still maintaining the main focus of the article.
  • In the introduction, we have clarified the objectives of the study and provided more context on the importance of understanding genetic diversity, population structure, and selection signatures in goats, particularly in relation to the Begait breed.
  • We have expanded the discussion on the methods and materials used in the study, including the selection of samples, whole-genome sequencing, and the analysis of genetic diversity, population structure, and selection signatures.
  • In the results section, we have presented the findings in a clear and organized manner, highlighting the key results and their implications for the Begait goat breed.
  • We have also discussed the limitations of the study and provided suggestions for future research to address these limitations.
  • In the conclusion, we have summarized the main findings of the study and their potential impact on the understanding of genetic diversity, population structure, and selection signatures in the Begait goat breed.

We believe that these revisions have improved the overall quality of the article and addressed the concerns raised in your feedback. We hope that the revised manuscript meets your expectations.

We thank you again for your big concern to improve the quality of the article.

Reviewer 2 Report

Comments and Suggestions for Authors

This manuscript describes genetic and genomic characteristics of Ethiopian goats. Such works are of great importance as they might be helpful in programs for conservation of local resources. The standard methods were used to describe genetic diversity and structure.

 However, I suggest that the manuscript needs revisions before consideration for publication.

In Introduction: for the statement “Until recently, only a few studies…” the citations are needed.

In Introduction: The part starting from “In this study, we discovered specific genes…” should be rewritten and given as the aim of this study. Here the authors should focus on what was the aim of their study, but not on the results.

In Material and Methods: “… from 5 unrelated goats from different breeds …” – It is unclear what are these 5 unrelated  animals. Did the authors mean five breeds or 44 individuals? The sentence needs to be corrected.  

In Material and Methods: …jaguar vine… - did the authors mean – jugular vein?

In Table 1, the full names of breeds should be given. At least in the notes (under the table). And if the authors focus on Begait goats, this breed should be first in the table.

In Methods: 2.2. The software Picard was used … to remove duplicates but not MarkDuplicates.

Why both GATK and SAMtools were used for SNP calling? The output from which software was used for further analysis?

In Methods 2.3: It is more correctly to write that Pn is proportion of polymorphic loci within a breed. SNPs are polymorphic loci, so the expression “polymorphic SNP” is incorrect

Which R package was used to plot PCA? I suggest it was made using standard R commands, without using any R packages.

In Methods 2.6: It is not clear what was used to annotate SNPs. Which caprine gene?

In Results: It is unclear what the authors meant by “we generated a total of 15,204,507,652 SNPs…”. How the number of SNPs can be larger than the length of the reference genome?

“… 16,439,441 SNPs were found to be polymorphic” – substitute “SNPs” with “loci”.

Table 2: the sign % is not needed here.  

For the ADMIXTURE analysis it would be more correctly to start from K=1. If starting from K=2, you cannot be sure that the minimum cross validation error was at 2 and not at 1.  

According to the phylogenetic tree (Figure 2a) there were several close relatives in Meafure (MR51011 and MR51012) and Begait (BG31026 and BG31027; BG31028 and BG31034; etc) breeds. And this could have affected the results of ADMIXTURE and PCA analysis (Figure 2c). I suggest that if the authors remove one of the pair of the relatives then the PCA and ADMIXTURE results will change. I believe that at least all the samples of Meafure will be in the same cluster.

To perform the studies on detection of selection signatures based on FST the studied groups should be presented with samples that 100% belong to their clusters. Here, based on PCA and ADMIXTURE analysis I do not see solid clusters to be compared. Moreover the filtering of loci that are not in Hardy Weinberg equilibrium should be performed for each breed separately. If it is performed for all the breeds at once then the fixed loci are removed (Wahlund effect). For example if a locus in one breed is AA in all the samples and TT in all the samples of another breed, and there are no heterozygotes then this locus will be removed due to HWE. But it could be a fixed locus with FST = 1.

In my opinion, the authors could substitute the section on detection of selection signatures, with Runs of Homozygosity and focus more on population structure and genetic diversity of Ethiopian breeds. Perhaps it would be interesting to include to the dataset some worldwide breeds that are available in ENA, NCBI, etc. and investigate relationships of breeds from Ethiopia with them. The detection of selection signatures could be performed as a different work, using more methods (like HapFLK, XP-EHH, etc) and using worldwide breeds to understand which genes are involved in adaptation to the Ethiopian conditions. And here I would like to underline that it is only my opinion and it is up to the authors to decide which sections to include.

Author Response

(The authors gave the same response as above.)

Reviewer 3 Report

Comments and Suggestions for Authors

Authors of the manuscript describes the study of the selection signatures of local Begait Goats. The study of the genetic diversity of local breeds, usually well adapted to the harsh climate environment is an interesting and important task. In this regard, I believe that this work provides a theoretical platform for a better understanding of the complex genomic architecture picture of the studied breeds. At the same time, there are a number of questions to the authors:

Some BG samples are located relatively close to other breeds. What could be the reason of this situation? Could you add a figure with projections PCA 1-3 for a better understanding of the relative location of the breeds? According to Fig 2b (Admixture), some Begait sample are close to AF, MR, CH populations.

Name of location in the Fig 1 is “Abeala”, in the Table 1 is “Abala”. Is this the same region or the different regions?

It is not entirely clear what is indicated in Figure 1 by the different colors. Is it enough to highlight the Tigray and Abala (Abeala?) regions, because samples are from these regions ?(According to the table 1)

Fig 5. “We found that the top regions are located at 11.9 - 11.20 Mb on chromosome 2…, 25.20- 25.40 Mb on chromosome 5…, 16.50 -16.70 Mb on chromosome 13…” In Figure 5, arrows indicate chromosomes 2, 5 and 24.

What about regions on chromosomes 3, 7, 13, 20?

Author Response

(The authors gave the same response as above.)
